# Drug- and Cell-Type-Specific Effects of ROCK Inhibitors as a Potential Cause of Reticular Corneal Epithelial Edema

**DOI:** 10.3390/cells14040258

**Published:** 2025-02-11

**Authors:** Ursula Schlötzer-Schrehardt, Andreas Gießl, Matthias Zenkel, Alexander Bartsch, Naoki Okumura, Noriko Koizumi, Shigeru Kinoshita, Theofilos Tourtas, Friedrich E. Kruse

**Affiliations:** 1Department of Ophthalmology, University Hospital Erlangen, Friedrich-Alexander University of Erlangen-Nürnberg, 91054 Erlangen, Germany; andreas.giessl@uk-erlangen.de (A.G.); matthias.zenkel@uk-erlangen.de (M.Z.); alexander.bartsch@uk-erlangen.de (A.B.); theofilos.tourtas@uk-erlangen.de (T.T.); friedrich.kruse@uk-erlangen.de (F.E.K.); 2Department of Biomedical Engineering, Faculty of Life and Medical Sciences, Doshisha University, Kyotanabe 610-0321, Japan; okumura.n@gmail.com (N.O.); noriko.k.koizumi@gmail.com (N.K.); 3Department of Frontier Medical Science and Technology for Ophthalmology, Kyoto Prefectural University of Medicine, Kyoto 602-0841, Japan; soy.skinoshi@gmail.com

**Keywords:** reticular bullous corneal epithelial edema, honeycomb epithelial edema, Rho kinase inhibitor, ROCK inhibitor, ripasudil, netarsudil, corneal epithelium, corneal endothelium

## Abstract

Rho-associated kinase (ROCK) inhibitors have gained popularity as novel treatment options in the management of glaucoma and corneal endothelial disorders. Among the various ocular side effects, reticular corneal epithelial edema has been most frequently reported, mainly after treatment with netarsudil. To explain the potential mechanisms, we comparatively analyzed the effects of ripasudil and netarsudil on corneal endothelial and epithelial function in vitro. Primary human corneal endothelial and epithelial cells were incubated with netarsudil dihydrochloride and ripasudil hydrochloride dihydrate for up to 7 days. Gene and protein expression analyses were performed by real-time PCR and immunocytochemistry. Functional assays assessed the cell migration, proliferation, viability, Na^+^/K^+^-ATPase activity, transcellular electrical resistance, and FITC–dextran permeability. Reticular bullous corneal epithelial edema was observed in a patient following netarsudil 0.02%/latanoprost 0.005% ophthalmic solution (Roclanda^®^) for elevated intraocular pressure. In the subsequent laboratory analyses, both netarsudil and ripasudil were found to improve the corneal endothelial pump and barrier function, but they showed differential effects on corneal epithelial cells. Whereas ripasudil improved the epithelial barrier function by upregulating major components of the tight and adherens junctions and reducing paracellular permeability, netarsudil had no or even adverse effects on the epithelial barrier properties by downregulating the expression levels of cell-junction-associated genes. The expression changes normalized after discontinuation of ROCK inhibitors. The findings support the concept that ROCK inhibitors can act as a double-edged sword by having beneficial effects on corneal endothelial cells and adverse effects on epithelial cells.

## 1. Introduction

Rho-associated kinase (ROCK) inhibitors are a novel class of anti-glaucomatous drugs, which lower the intraocular pressure (IOP) mainly by reducing the resistance to the aqueous humor outflow through the trabecular meshwork by disrupting actin stress fibers [1]. Two ROCK inhibitors are commercially available for the treatment of open-angle glaucoma and ocular hypertension: the ROCK-I and ROCK-II inhibitor ripasudil (Glanatec^®^ ophthalmic solution 0.4%; Kowa Company Ltd., Nagoya, Japan), approved in 2014 for use in Japan [2], and the ROCK and norepinephrine transporter inhibitor netarsudil (Rhopressa^®^ ophthalmic solution 0.02%; Aerie Pharmaceuticals, Durham, NC, USA), approved by the US Food and Drug Administration (FDA) in 2017 for use in the United States [3]. A fixed-dose combination of netarsudil 0.02% and latanoprost 0.005%, a prostaglandin analog, was later approved in the United States and in Europe (Rocklatan^®^ and Roclanda^®^, respectively; Aerie Pharmaceuticals) [4]. Both ripasudil and netarsudil have demonstrated a good safety profile, with only minimal side effects in clinical trials [5,6]. Nevertheless, the clinical properties of ripasudil were found to be different from those of netarsudil, particularly relating to the different side effects in the cornea and conjunctiva [7,8,9]. The most common adverse effects of ripasudil were conjunctival hyperemia, blepharitis and ocular irritation, whereas subconjunctival hemorrhage, conjunctival edema and cornea verticillata were mostly limited to netarsudil. Recently, a possible association between long-term netarsudil use and crystalline keratopathy has been reported in a glaucoma patient [10]. Although not reported in larger clinical trials, single cases of netarsudil-induced reticular epithelial edema were reported as an additional potential side effect in both adult and pediatric glaucoma patients with or without pre-existing corneal disease [11,12,13,14,15,16,17].

These differences in the safety profiles were also seen during off-label use of ROCK inhibitors for the management of corneal endothelial dysfunction, such as Fuchs endothelial corneal dystrophy (FECD) [18,19,20] and corneal endothelial restoration in surgical interventions and cell-based therapies [21,22,23]. In particular, the use of both ripasudil and netarsudil after Descemet stripping only (DSO) procedures in patients with FECD has been expanding rapidly to promote peripheral endothelial cell proliferation and/or migration after surgery, with a success rate of corneal clearance in 85% to 90% of cases [24,25,26]. However, an increasing number of case reports showed episodes of reticular bullous epithelial edema occurring mainly with the use of netarsudil and, to a much lesser extent, with ripasudil within weeks to months [13,14,17,27,28,29]. This complication was reported mainly in patients with a prior history of endothelial compromise and corneal edema, previous ocular surgery or ocular comorbidities, such as uveitis and glaucoma predisposing to corneal edema, with an associated decline in vision within several weeks of exposure to the medication. However, the reticular corneal epithelial edema usually resolved upon discontinuation of the medication [13,14,17,27,28,29].

Although the mechanism of epithelial bullae development is unclear, hypotheses suggested either a potential shift of stromal edema to the cornea epithelium due to improved endothelial pump function or direct drug effects on corneal epithelial intercellular junctions and barrier properties [13,19]. In order to better understand the relative benefits and risks of both drugs for corneal cells and to explain the potential mechanisms underlying reticular epithelial edema, we tested both hypotheses with an in vitro approach. To this end, we comparatively analyzed the effects of ripasudil and netarsudil on the gene expression and function of corneal epithelial and endothelial cells in vitro.

## 2. Methods

### 2.1. Human Tissues and Study Approval

Normal donor corneas (*n* = 10) unsuitable for transplantation, with appropriate research consent, were obtained from the Cornea Bank Erlangen. The mean donor age was 58.2 ± 10.3 years (7 male, 3 female).

Central endothelial cell–Descemet membrane (EDM) lamellae, 8 mm in diameter, were prepared by manual stripping from FECD patients (*n* = 10) during Descemet membrane endothelial keratoplasty (DMEK) surgery. The mean patient age was 69.7 ± 9.6 years (5 male, 5 female). Immediately after excision, the EDM scrolls were dissected into halves and incubated for 24–72 h in 4 mL of CorneaMax storage medium (Eurobio, Les Ulis, France) without or with a single dose of either ripasudil hydrochloride dihydrate (K-115) or netarsudil dihydrochloride (AR-13324) (Selleck Chemicals, Houston, TX, USA) at 37 °C. Following incubation, the paired specimens were subjected to immunohistochemistry or functional assays.

The handling of the donor tissues was in accordance with the principles of the Declaration of Helsinki for experiments involving human tissues and samples. Ethics approval for this laboratory investigation was obtained from the Institutional Review Board of the Medical Faculty of the University of Erlangen-Nürnberg (No. 138_18B, 24 April 2018), and informed research consent was obtained from all the patients, including the clinical case.

### 2.2. Cell Culture

Primary human corneolimbal epithelial cells (pHCEpC) were prepared from organ-cultured donor corneas (*n* = 10) as previously described [30] and cultured in keratinocyte serum-free medium (KSFM; Thermo Fisher Scientific; Schwerte, Germany) containing 25 µg/mL bovine pituitary extract (BPE) and 5 ng/mL human epidermal growth factor (EGF). The cells were characterized by the expression of KRT3 (keratin3), KRT12 (keratin 12) and TP63 (tumor protein 63) on the mRNA and protein levels.

Human primary corneal endothelial cells (pHCEnC) were obtained from Celprogen (Torrance, CA), cultured in Opti-MEM I (Thermo Fisher Scientific) containing 8% fetal calf serum (FCS), 200 µg/mL CaCl_2_, 20 µg/mL ascorbic acid, 0.08% chondroitin sulfate, 5 ng/mL EGF (Abcam; Cambridge, UK) and 50 µg/mL gentamycin on precoated (iMatrix-511; Takara Bio, Shiga, Japan) cell culture ware and tested for the expression of endothelial cell markers, such as Na^+^/K^+^-ATPase, ZO-1 and N-cadherin.

Human Tenon’s capsule fibroblast (hTCF) cultures were established as previously described [31] and maintained in Dulbecco’s modified Eagle medium (DMEM) containing 15% FCS and 1% penicillin/streptomycin/amphotericin B (PSA) solution (PAN-Biotech, Aidenbach, Germany).

Epithelial and endothelial cells were incubated with 1 µM netarsudil dihydrochloride (AR-13324) and 10 µM and 30 µM ripasudil hydrochloride dihydrate (K-115) (Selleck Chemicals, Houston, TX, USA) for up to 7 days according to the published literature [32,33] and our previous studies [34]. To test the effects of ripasudil and netarsudil, the cells were incubated in serum-deprived medium (pHCEpC: 0% FCS, pHCEnC: 0.05% FCS, hTCF: 1% FCS, respectively) for 6 to 96 h.

### 2.3. Functional Cell Assays

The cell viability was assessed using a two-color fluorescence staining kit (Life/Dead Viability/Cytotoxicity kit; Molecular Probes, Eugene, OR, USA) and a plate reader (Fluoroskan Ascent FL 2.4; Thermo Scientific, Bonn, Germany), as recommended by the manufacturer.

Cell migration assays were performed as described previously [30]. Briefly, 1 × 10^5^ pHCEpC/mL were seeded into 2-well culture inserts with a defined cell-free gap (ibidi GmbH, Planegg, Germany) and cultivated overnight. After formation of a cellular monolayer, the silicone inserts were removed and the cells were incubated without and with ROCK inhibitor. Images of each well were acquired immediately following insert removal (0 h) and after 4, 7 and 10 h by using an inverted microscope (BX51; Olympus, Hamburg, Germany). The image analysis software ZEN version 1.1.2.0 (Carl Zeiss Microscopy, Oberkochen, Germany) was used to measure the changes in the cell-covered areas over time. All the experiments were performed in triplicate.

The cell proliferation was analyzed using the Cell Proliferation ELISA BrdU Colorimetric Assay Kit (Roche Diagnostics, Mannheim, Germany) as described previously [30]. Briefly, pHCEpCs were seeded into 24-well plates at a density of 10,000 cells/well and cultured in serum-reduced medium for 24 h without or with ROCK inhibitor, and they were labeled with BrdU according to the manufacturer’s instructions. The absorbance was measured at 450 nm using a spectrophotometer (Multiskan Spectrum; Thermo Scientific, Waltham, MA, USA), and the fold change values were calculated to the normalized control. The experiments were performed in triplicates.

The Na^+^/K^+^-ATPase activity was measured using a colorimetric ATPase assay kit (Abcam; Cambridge, UK). Cultured cells and EDM specimens were incubated without and with ROCK inhibitor for 48 h. The cells were lysed in ice-cold ATPase assay buffer and treated according to the manufacturer’s instructions. The absorbance was measured at 650 nm in endpoint mode using a spectrophotometer (Multiskan Spectrum) and the enzymatic activity was quantified by using a standard curve.

Transendothelial/epithelial electrical resistance (TEER) measurements were performed on confluent monolayer cultures of pHCEpCs, pHCEnCs and fibroblasts seeded on Transwell culture inserts (1.0 µm pore size, PET membrane; Greiner Bio-One, Frickenhausen, Germany) in 24-well plates. The cells were incubated in serum-reduced medium without and with ROCK inhibitor for 48 h. The measurements were performed with an EVOM3 (Epithelial Volt Ohm Meter; World Precision Instruments, Friedberg, Germany) and calculated according to the company’s instructions.

Fluorescein isothiocyanate (FITC)–dextran permeability measurements were performed on confluent monolayer cultures of pHCEpCs, pHCEnCs and hTCFs seeded on Transwell culture inserts (1.0 µm pore size, PET membrane; Greiner Bio-One) in 24-well plates. The cells were incubated in serum-deprived medium without and with ROCK inhibitor for 48 h. Subsequently, the medium in the top wells was replaced by 0.5 µg/mL of FITC–dextran (molecular weight 10 kDa) (Sigma-Aldrich; St. Louis, MO, USA) in PBS and the medium in the bottom wells was replaced by PBS. The concentrations of the FITC–dextran in the bottom well were measured after 2 and 4 h using a plate reader (Fluoroskan Ascent FL 2.4) at an excitation wavelength of 490 nm and an emission wavelength of 530 nm. The fluorescence intensity of the control medium was used as the background control.

### 2.4. Real-Time RT-PCR

RNA isolation from the EDM specimens and cultured cells was performed using the RNeasy Micro Kit (Qiagen, Hilden, Germany) or the RNeasy Mini Kit, respectively, including an on-column DNase digestion step. First-strand cDNA synthesis was performed using 50 ng of RNA from the EDM specimens or 1 µg of RNA from the cultured cells, 200 U Superscript II reverse transcriptase and 200 ng random primers (Thermo Fisher Scientific) in a 20 µL reaction volume. Quantitative real-time PCR was performed using the CFX Connect thermal cycler and software (latest v. 2.3, Bio-Rad, Munich, Germany). The PCR reactions were run in duplicate and contained diluted first-strand cDNA, 0.4 µM each of upstream and downstream primer, and 1x SsoFast EvaGreen Supermix (Bio-Rad), respectively, according to the manufacturers’ recommendations. The primer sequences (Eurofins; Anzing, Germany) designed using Primer 3 software and the reaction conditions are given in Appendix A. The expression levels were normalized to the housekeeping genes GAPDH, HPRT1, and RPLP0.

### 2.5. Immunocytochemistry

Cell cultures were fixed in 4% paraformaldehyde in phosphate-buffered saline (PBS) for 15 min, permeabilized with 0.5% Triton X-100 in PBS for 10 min, blocked with 10% normal goat serum for 30 min, and incubated in the primary antibodies (Appendix A) diluted in PBS overnight at 4 °C. The F-actin filaments were labeled with Alexa Fluor 555-Phalloidin (Cell Signaling Technology; Danvers, MA, USA). Antibody binding was detected by the Alexa-conjugated secondary antibodies (Molecular Probes; Eugene, OR, USA) and nuclear counterstaining was performed with 4′,6-diamidino-2-phenylindole (DAPI; Sigma-Aldrich). The immunolabeled tissues were radially incised and flat mounted in Aqua-Poly/Mount (Polysciences; Warrington, PA, USA) embedding medium before being examined with a laser scanning confocal microscope (LSM 780; Carl Zeiss Microscopy). In the negative control experiments, the primary antibodies were replaced by equimolar concentrations of an irrelevant isotypic primary antibody.

### 2.6. Statistical Analysis

Statistical analyses were performed using GraphPad Prism V9.5.1 (GraphPad Software, San Diego, CA, USA) software. Data are expressed as the mean ± standard deviation (SD). The Gaussian distribution of the datasets was tested using the Kolmogorov–Smirnov or Shapiro–Wilk normality tests. The significance of the differences between groups was calculated by Student’s *t*-test or the Mann–Whitney *U*-test. A *p* value of <0.05 was considered statistically significant.

## 3. Results

### 3.1. Clinical Case Report

A 32-year-old female with a history of congenital hereditary endothelial dystrophy (CHED) and secondary open-angle glaucoma presented with an elevated IOP of 23 mm Hg (Goldmann applanation tonometry) in her left eye. She had undergone multiple intraocular surgeries, including penetrating keratoplasties (1996, 2010, 2019 and 2021), trabeculectomy (2012) and cyclophotocoagulation (2022). At presentation, the patient was using Clonid Ophthal 1/8% sine (clonidine hydrochloride 0.125%), Timo EDO 0.5% (timolol 0.5%), and Dexa EDO (dexamethasone 0.1%) as local treatment. Roclanda^®^ (netarsudil 0.02%/latanoprost 0.005% ophthalmic solution) was added for IOP control in April 2023. Six months later, in October 2023, she presented with reticular corneal epithelial edema in the inferior cornea (Figure 1), likely due to netarsudil, which completely resolved 4 weeks after discontinuation of the Roclanda^®^ medication. The development of reticular corneal epithelial edema following a fixed-dose combination of netarsudil and latanoprost has previously not been reported in the literature. However, this observation initiated an in vitro investigation into the differential effects of netarsudil and ripasudil on corneal endothelial and epithelial cells by testing two major hypotheses regarding the mechanisms underlying these clinical findings [13].

### 3.2. Effects of Ripasudil and Netarsudil on Corneal Endothelial Pump Function

To test the hypothesis that improved endothelial pump function causes a shift from stromal edema to the corneal epithelium, we first comparatively analyzed the effects of ripasudil and netarsudil on endothelial pump function in vitro. We used primary human corneal endothelial cells (pHCEnCs), which were characterized by positive staining for Na^+^/K^+^-ATPase, ZO-1, and N-cadherin [35] (Figure 2A) and which were incubated with commonly used experimental concentrations of ripasudil (10 and 30 µM) and netarsudil (1 µM), corresponding to 0.0004% and 0.0012% ripasudil and 0.00005% netarsudil, for 72 h [32,33,34]. At these subclinical concentrations, both drugs had no adverse effects on the cell phenotype and viability (Figure 2B). As expected, both ROCK inhibitors significantly increased the cell proliferation and migration compared with untreated control cells (Figure 2C,D; Appendix A).

The effect of the ROCK inhibitors on the corneal endothelial pump function was determined by analyzing the expression levels of key genes important for ion and water transport as previously described [34]. Both ripasudil and netarsudil were found to upregulate the expression levels of the Na^+^/K^+^-ATPase subunits *ATP1A1* and *ATP1B1*, bicarbonate transporters *SLC4* (solute carrier family 4) *A4*, *SLC4A7* and *SLC4A11*, monocarboxylate transporters *SLC16A1* and *SLC16A7*, Na^+^/H^+^ antiporter *SLC9A1*, aquaporin *AQP11* and carbonic anhydrase *CA2* at 24–72 h of incubation compared to the untreated controls (Figure 3A; Appendix A). Whereas the expression of the ATPase subunits *ATP1B3* and *ATP2A1*, chloride transporter *SLC12A2,* and *AQP1* was upregulated by ripasudil only, sialic acid transporter *SLC17A5* was induced by netarsudil. The expression levels of *ATP2B1* and *SLC16A3* were not altered by both drugs, and *ATP1A4*, *SLC7A14* and *SLC12A1* were not expressed in pHCEnCs at all.

Immunocytochemistry and functional assays provided further evidence that the Na^+^/K^+^-ATPase protein expression and enzymatic activity, indicating enhanced pump function, was significantly increased in the ripasudil- and netarsudil-treated pHCEnCs without any significant differences (Figure 3B). In addition, the EDM lamellae obtained from FECD patients during DMEK surgery showed a significant increase in Na^+^/K^+^-ATPase protein expression and Na^+^/K^+^-ATPase activity after incubation in both 30 µM ripasudil and 1µM netarsudil, confirming the in vitro studies (Figure 3B).

### 3.3. Effects of Ripasudil and Netarsudil on Corneal Endothelial Barrier Function

Expression of the tight-junction-associated genes *TJP1* (ZO-1) and *TJP2* (ZO-2) was upregulated by both ripasudil and netarsudil, whereas *CLDN2* (claudin-2) and *OCLN* (occludin) were upregulated by ripasudil only (Figure 4A). Similarly, the gap-junction-associated gene *GJA1* (connexin 43) and the adherens-junction-associated genes *CDH2* (N-cadherin), *ITGA6* (integrin α6) and *ITGB1* (integrin ß1) were upregulated by ripasudil only.

Although ripasudil appeared to exceed the regulatory activity of netarsudil, immunocytochemistry and functional assays provided evidence that the ZO-1 protein expression and barrier function were significantly increased in the ripasudil- and netarsudil-treated pHCEnC monolayers compared to the controls, without any significant differences between both drugs (Figure 4B). Specifically, fluorescein isothiocyanate (FITC)–dextran permeability assays showed the reduced tight-junction-related, paracellular permeability of confluent cell layers upon ripasudil and netarsudil incubation compared to the untreated pHCEnCs and fibroblasts as controls (Figure 4B). Consistently, transendothelial electrical resistance (TEER) assays revealed a significant increase in ohmic resistance following ROCK inhibition compared to the controls (Figure 4B).

Taken together, these data indicate that both ROCK inhibitors improve endothelial pump and barrier function in a comparable manner, thereby disproving the hypothesis and supporting the equal use of both ROCK inhibitors in the management of corneal endothelial dysfunction.

### 3.4. Effects of Ripasudil and Netarsudil on Corneal Epithelial Barrier Function

To test the second hypothesis that ROCK inhibitors directly affect corneal epithelial tight junctions and paracellular permeability, we comparatively analyzed the effects of ripasudil and netarsudil on the cell junctions and barrier properties of corneal epithelial cells. We used primary human corneal endothelial cells (pHCEpCs), which were characterized by the expression of *KRT3* (keratin 3), *KRT12* (keratin 12) and *TP63* (tumor protein p63 alpha) (Appendix A) and which were incubated with ripasudil (10 and 30 µM) and netarsudil (1 µM). We first analyzed the expression levels of key genes involved in cell–cell and cell–matrix adhesion, which form a prerequisite for the protective barrier function of the epithelium. The expression levels of the adherens-junction-associated gene *CDH1* (E-cadherin), the desmosome-associated genes *DSG1* (desmoglein 1) and *DSC2* (desmocollin 2) as well as the hemidesmosomal genes *PLEC* (plectin), *ITGA6* (integrin α6) and *ITGB4* (integrin ß4) were consistently upregulated by ripasudil throughout 7 days, but they were notably not affected or even downregulated by netarsudil, as was particularly evident for the reduced expression of *ITGA6* and *ITGB4* up to 5 days of incubation (Figure 5A). Similar findings were obtained for the tight-junction-associated genes *TJP1* (ZO-1), *OCLN* (occludin) and *CLDN1* (claudin-1), which were consistently upregulated by ripasudil at both concentrations but were either not affected or partly downregulated by netarsudil (Figure 5A). Only the expression levels of the gap junction marker *GJA1* (connexin 43) and the aquaporins *AQP3* and *AQP5* were significantly upregulated by both drugs compared to the untreated control cells (Figure 5A).

The differential upregulation of cell adhesion molecules by the ROCK inhibitors could also be confirmed on the protein level using immunocytochemistry, showing markedly reduced staining for E-cadherin, integrin α6 and ZO-1 upon exposure to netarsudil compared to ripasudil (Figure 5B). Moreover, the FITC–dextran permeability assay showed the reduced tight-junction-related paracellular permeability of the confluent epithelial cell layers upon incubation with ripasudil (10µM and 30 µM) compared to untreated pHCEpCs and fibroblast controls, whereas no significant effect was observed following incubation with netarsudil (Figure 5C). However, TEER measurements revealed a significant increase in transepithelial resistance, which is tight-junction-independent, following incubation with both ripasudil and netarsudil compared to the controls (Figure 5C).

Discontinuation of ROCK inhibitor treatment after 3 days resulted in the normalization of the expression alterations by both ripasudil and netarsudil to control levels after 6 days, indicating the reversibility of the expression changes and confirming the clinical observations (Figure 6).

Taken together, these data indicate that the two ROCK inhibitors affect corneal epithelial cells differently, with ripasudil being slightly superior to netarsudil in improving epithelial barrier function and resistance to fluid entry, obviously confirming this hypothesis.

### 3.5. Effects of Ripasudil and Netarsudil on Corneal Epithelial Actin Cytoskeleton

In addition to the transcriptional alterations, ROCK inhibitors may also regulate the assembly of cell junctions indirectly via their known effects on the F-actin cytoskeleton, which is firmly linked to the tight junction and adherens junction proteins [36]. As expected, both ROCK inhibitors induced the reorganization and disassembly of epithelial F-actin stress fibers, stained with phalloidin, into a scattered, dispersed F-actin belt in the cell periphery, which resulted in phenotypic changes, including the relaxation, enlargement and rounding of treated cells (Figure 7). However, the increased formation of ZO-1- and occludin-positive tight junctions was only observed after ripasudil treatment (Figure 7A). Similarly, the increased formation and stabilization of cadherin-1-positive adherens junctions were mainly observed after incubation with ripasudil (Figure 7B).

Thus, the indirect effects of the ROCK inhibitors on cell junction formation and stabilization via the cytoskeleton may also contribute to their differential impact on corneal epithelial cells.

## 4. Discussion

In recent years, topical ROCK inhibitors, including ripasudil and netarsudil, have gained considerable popularity in ophthalmology, particularly in the management of ocular hypertension and glaucoma, diabetic retinopathy and corneal endothelial disorders [37]. In clinical trials, these novel drugs have shown efficacy and good safety profiles, with only minimal side effects. However, due to the pleiotropic functions and ubiquitous regulatory roles of the Rho/ROCK signaling pathway, ROCK inhibition has the potential to cause unintended ocular side effects, which have not been reported in these clinical trials [9]. These include corneal neovascularization and hemorrhages, anterior subcapsular lens opacities, punctal stenosis, corneal flattening and crystalline keratopathy, which have been solely described after netarsudil treatment [10]. The most widely reported side effect of topical ROCK pathway inhibition with netarsudil has been reticular corneal epithelial edema, which has been described in numerous case reports and case series [11,12,13,14,15,16,17,28]. The majority of patients had a history of corneal endothelial dysfunction and corneal edema, surgical procedures or ocular comorbidities, including uveitis and glaucoma, predisposing them to corneal edema. In these cases, the adverse effects of netarsudil on corneal epithelial cells may override the beneficial effects of netarsudil on corneal endothelial cells [21,22,23,24,33,38,39]. Despite several reasonable hypotheses on the development of epithelial microcysts in predisposed patients, a high degree of uncertainty remains regarding the differential effects of specific ROCK inhibitors on corneal cells. In view of the growing popularity for both on- and off-label use of ROCK inhibitors in ophthalmology, improved understanding of the mechanisms underlying the potential side effects is required.

This study not only reported the first case of reticular epithelial edema following a fixed combination of netarsudil and latanoprost (Roclanda^®^) for IOP control but also provided a detailed laboratory analysis of the differential effects of netarsudil and ripasudil on primary corneal endothelial and epithelial cells. In order to test the two major hypotheses explaining the alterations in fluid dynamics in the cornea [13], we comparatively analyzed the ROCK inhibitors’ effects on corneal endothelial pump function and corneal epithelial barrier properties in vitro.

### 4.1. ROCK Inhibitor Effects on Corneal Endothelial Pump Function

The findings of this study provided further evidence that ROCK inhibitors have the potential to improve endothelial pump and barrier function, even in the diseased endothelial cells of FECD patients [34]. In this respect, ripasudil and netarsudil showed no significant differences, as indicated by equally upregulated expression levels of key genes important for ion and water transport, specifically sodium–potassium pump subunits, bicarbonate transporters, monocarboxylate transporters, and chloride transporters, and by the enhanced Na^+^/K^+^-ATPase enzyme activity in both primary cell cultures and surgically excised EDM specimens from FECD patients. These data appear to refute the hypothesis that netarsudil shifts the pre-existing stromal edema to the corneal epithelium due to increased endothelial pump function. On the other hand, the findings substantiate the equal use of both ROCK inhibitors in the clinical management of corneal endothelial dysfunction, such as FECD, as well as in corneal endothelial regeneration and/or protection in surgical interventions, such as cataract surgery and DSO [19,20,40,41].

### 4.2. ROCK Inhibitor Effects on Corneal Epithelial Cell Junctions and Barrier Function

The establishment and maintenance of cell–cell and cell–matrix junctions is crucially important to regulate the adhesion, apico-basal polarity and barrier properties of epithelial and endothelial cells. Adherens junctions, including desmosomes and hemidesmosomes, mediate cell–cell or cell–matrix adhesion, gap junctions enable communication and exchange between cells, and tight or occluding junctions provide an effective barrier [42]. Adherens and tight junctions are regulated by the actin cytoskeleton, and reorganization of the actin cytoskeleton by Rho kinases can either stabilize or disrupt the cell junctions in a cell-type-specific manner [43]. In general, activated Rho signaling induces the formation of stress fibers, thereby increasing intracellular tension and destabilizing cell junctions, leading to altered barrier properties [44]. Consistently, ROCK inhibitors have been reported to improve the barrier integrity in epithelial and endothelial cells by different molecular mechanisms, including reorganization of actin stress fibers and direct regulation/phosphorylation of tight junction components, including occludin and claudin [45,46]. In bovine corneal endothelial cells, activation of Rho signaling induced the contraction of the actin cytoskeleton and redistribution of ZO-1 and cadherins, leading to a decrease in transendothelial electrical resistance; these effects were reversed by the ROCK inhibitor Y-27632 [47].

In line with these notions, the findings of this study provided evidence that ripasudil improved both endothelial and epithelial barrier function by consistently upregulating the expression of the tight-junction-associated genes *TJP1* (ZO-1), *TJP2* (ZO-2), *CLDN1* (claudin-1), *CLDN2* (claudin-2) and *OCLN* (occludin), as well as the adherens-junction-associated genes *CDH1* (cadherin-1), *CDH2* (cadherin-2), *DSG1* (desmoglein 1), *DSC2* (desmocollin 2), *PLEC* (plectin), *ITGA6*, *ITGB1* and *ITGB4* (integrin α6, ß1 and ß4), in corneal endothelial and epithelial cells on both the mRNA and protein levels. Further evidence of improved barrier function with ripasudil was also provided by the reduction of paracellular permeability and enhancement of transcellular resistance of confluent endothelial and epithelial cell monolayers in functional assays. In addition to the direct transcriptional effects on junctional genes, ripasudil may also stimulate increased assembly of cell junctions indirectly via their known disruptive effects on actin stress fibers, leading to the cellular relaxation and stabilization of cell junctions in corneal epithelial cells [44,48].

In contrast, the expression levels of almost all tight-junction- and adherens-junction-associated genes in endothelial and epithelial cells were either not affected or even downregulated by netarsudil compared to the controls. Regarding corneal epithelial cells, the tight junction component occludin and the hemidesmosomal components integrin α6 and integrin ß4 were most significantly downregulated by netarsudil, potentially weakening the adhesion of epithelial cells among each other and to their basement membrane, thereby facilitating fluid entry from the stroma to the epithelium. In accordance, netarsudil, other than ripasudil, did not improve the tight-junction-dependent paracellular permeability of the confluent epithelial cell monolayers. Discontinuation of ROCK inhibitor treatment after 3 days resulted in the normalization of the expression alterations to control levels after 6 days, indicating the reversibility of the changes.

These differential drug- and cell-type-specific effects may be related to the different pharmacokinetic profiles of ripasudil and netarsudil [49]. Netarsudil, a ROCK and norepinephrine transporter inhibitor, is a cationic and amphiphilic drug, which is more lipophilic than ripasudil and exerts additional IOP-lowering mechanisms and side effects that have not been reported for other ROCK inhibitors [1]. Norepinephrine is a monoamine neurotransmitter, and monoamine receptors have been found in both the corneal epithelium and the endothelium, where they may play a role in homeostasis and fluid transport [50]. It is therefore conceivable that the difference in the incidence of reticular epithelial edema between netarsudil and ripasudil is associated with this key pharmacologic distinction.

This study has some limitations. A major limitation is that only a 2D in vitro model of corneal epithelial cells has been used to assess the effects of ROCK inhibitors on epithelial barrier function. To better recreate the multilayered in vivo tissue architecture, 3D in vitro models of the corneal epithelium should be used in future approaches, such as primary human corneal epithelial cells forming stratified epithelia at the air–liquid interface [51]. Using such models will substantiate and further improve analyses of drug effects on cellular interactions, barrier disruptions and permeability functions. Another limitation of the present study is the selective approach to gene expression alterations using appropriate markers of endothelial pump and barrier, as well as epithelial barrier, function. While this limited approach may have been sufficient to test existing hypotheses on the development of epithelial bullae in vitro, future studies should perform an unbiased transcriptomic analysis to provide further mechanistic insights into the drug-induced cellular and molecular effects and regulatory networks between differentially expressed genes.

## 5. Conclusions

The findings of this study provide evidence that the effects of ROCK inhibitors on corneal endothelial and epithelial cells are drug- and cell-type-specific. They also support the notion that ROCK inhibitors can act as a double-edged sword by improving corneal endothelial function and corneal stromal edema but also causing reticular epithelial edema in predisposed patients [21]. Whereas both netarsudil and ripasudil were found to improve corneal endothelial pump and barrier function compared to untreated controls, they showed differential effects on corneal epithelial cells, with ripasudil improving and netarsudil impairing epithelial barrier function. The occurrence of reticular bullous epithelial edema after netarsudil treatment may be caused by both direct but reversible transcriptional alterations of cell-junction-associated genes and indirect effects on paracellular permeability via the disruption of the actin cytoskeleton in epithelial cells, thereby facilitating fluid entry from the stroma in the presence of a compromised endothelium. As ROCK inhibitors continue to be increasingly used in different subspecialties, ophthalmologists and patients need to be aware of their potential benefits and side effects, together with their mechanisms of development.

## Figures and Tables

**Figure 1 cells-14-00258-f001:**
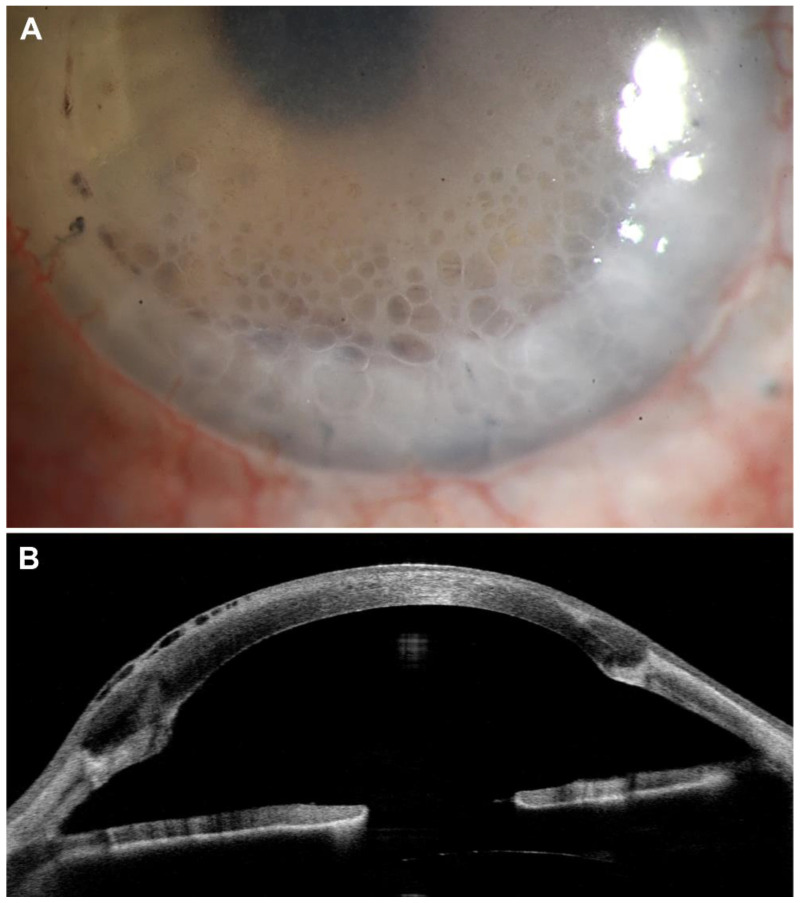
Reticular corneal epithelial edema in a 32-year-old female patient with a history of congenital hereditary endothelial dystrophy (CHED) and secondary open-angle glaucoma, who had received netarsudil 0.02%/latanoprost 0.005% ophthalmic solution (Roclanda^®^) for the treatment of elevated intraocular pressure. (**A**) Slit lamp biomicroscopy showing reticular epithelial cysts in the inferior cornea. (**B**) Anterior segment optical coherence tomography demonstrating epithelial bullae (magnification bar = 500 µm).

**Figure 2 cells-14-00258-f002:**
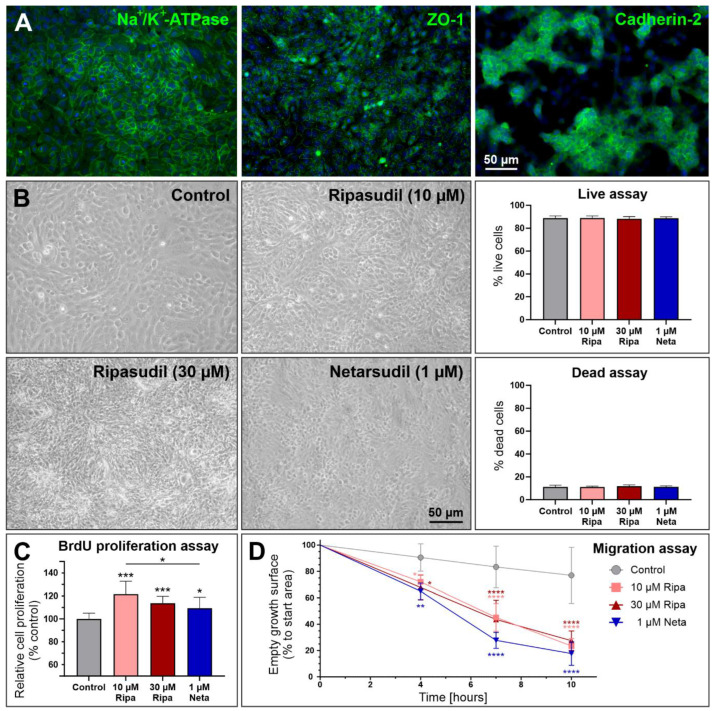
Characterization of primary human corneal endothelial cells (pHCEnCs) treated without or with ripasudil (10 and 30 µM) and netarsudil (1 µM), respectively. (**A**) Positive immunofluorescence staining for Na^+^/K^+^-ATPase, ZO-1 and cadherin-2. (**B**) Phase contrast microscopic images of pHCEnCs showing the cell phenotype and live/dead viability assays (*n* = 8) following incubation with ROCK inhibitors for 72 h. (**C**) Effects of ROCK inhibitors on pHCEnC proliferation as assessed by spectrophotometric measurement of BrdU incorporation 24 h after incubation (*n* = 9). (**D**) Effects of ROCK inhibitors on pHCEnC migration as assessed by measurement of the gap closure 10 h after removal of culture inserts (*n* = 3) (data are expressed as means ± SDs relative to controls set to 100%; * *p* < 0.05; ** *p* < 0.01; *** *p* < 0.001; **** *p* < 0.0001; Mann–Whitney *U* test).

**Figure 3 cells-14-00258-f003:**
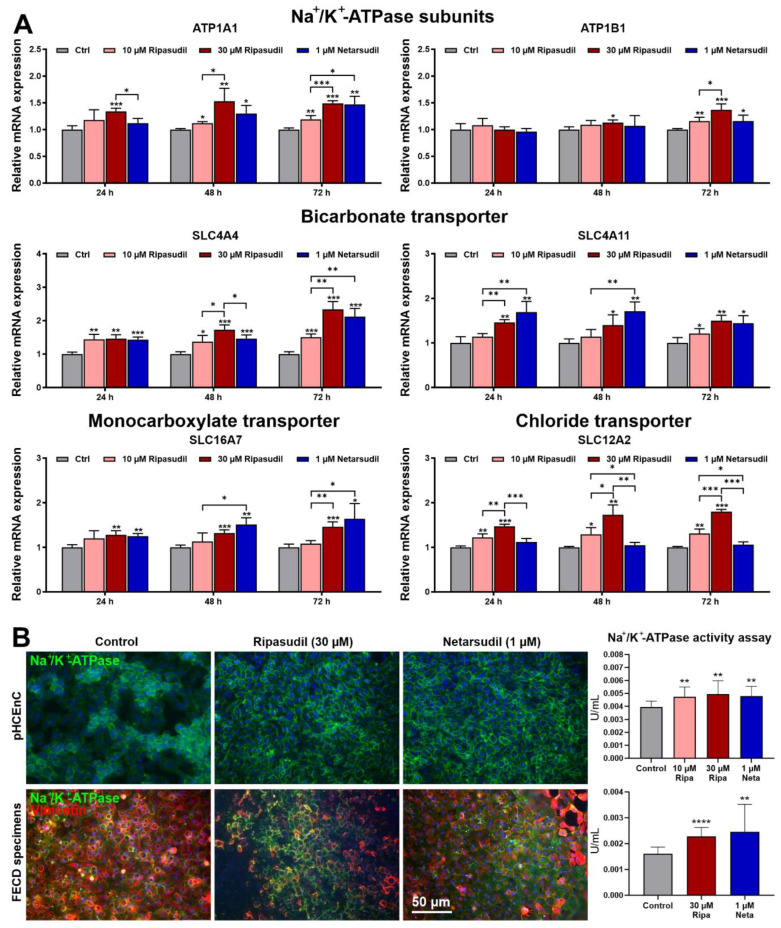
Effects of ripasudil and netarsudil on corneal endothelial pump function. (**A**) Quantitative real-time PCR analysis of primary human corneal endothelial cells (pHCEnCs) treated without or with ripasudil (10 and 30 µM) and netarsudil (1 µM) for 24 to 72 h, respectively (*n* = 6), showing the relative mRNA expression levels of *ATP1A1* (Na,K-ATPase alpha-1 subunit), *ATP1B1* (Na,K-ATPase beta-1 subunit), *SLC4A4* (solute carrier family 4 member 4), *SLC4A11* (solute carrier family 4 member 11), *SLC16A7* (solute carrier family 16 member 7), and *SLC12A2* (solute carrier family 12 member 2). Data are normalized to *GAPDH* (glyceraldehyde-3-phosphate dehydrogenase) and *HPRT1* (hypoxanthine phosphoribosyltransferase 1) and expressed as the means ±  SDs relative to controls set to 1 (* *p* < 0.05, ** *p* < 0.01,*** *p* < 0.001; unpaired *t*-test). (**B**) Expression of Na^+^/K^+^-ATPase (green fluorescence) and vimentin (red fluorescence) in pHCEnCs (top row) and endothelial cell–Descemet membrane (EDM) lamellae obtained from patients with Fuchs endothelial corneal dystrophy (FECD) (bottom row) incubated without or with ripasudil (30 µM) and netarsudil (1 µM) for 48 h (DAPI nuclear counterstain in blue). The Na^+^/K^+^-ATPase activity (*n* = 6) was also measured in pHCEnCs and EDM specimens following ROCK inhibitor incubation for 48 h (data are expressed as the means ± SDs relative to controls set to 100%; ** *p* < 0.01; **** *p* < 0.0001; Mann–Whitney *U* test).

**Figure 4 cells-14-00258-f004:**
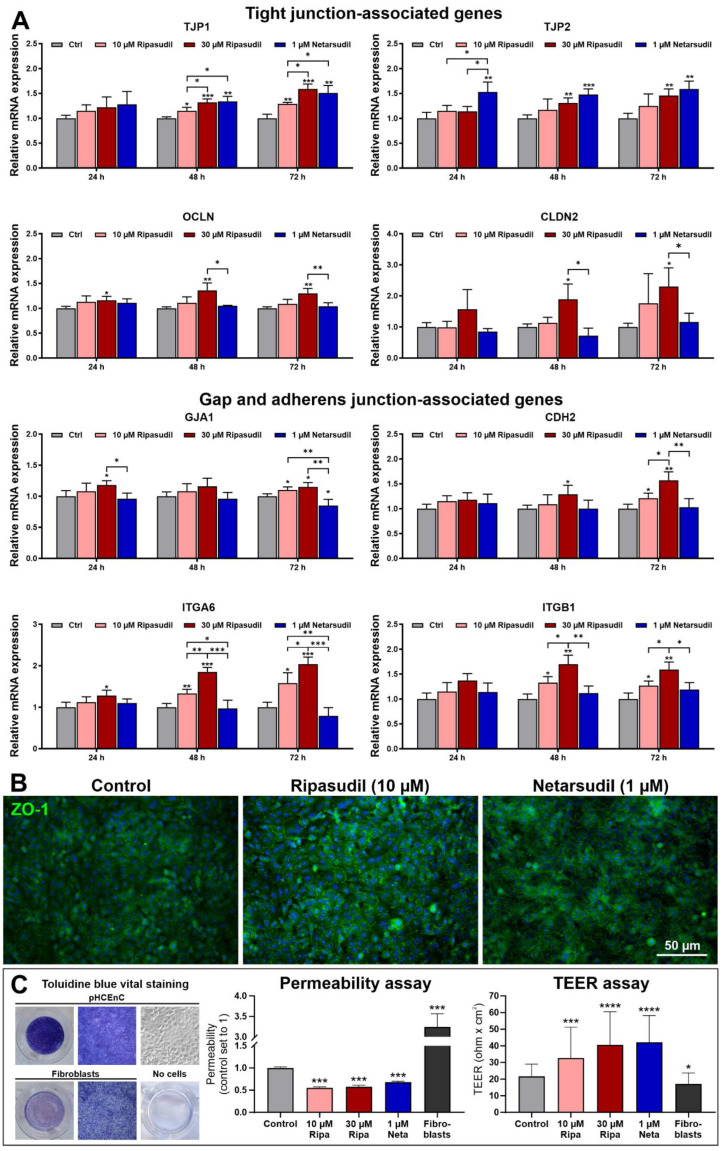
Effects of ripasudil and netarsudil on corneal endothelial pump function. (**A**) Quantitative real-time PCR analysis of primary human corneal endothelial cells (pHCEnCs) treated without or with ripasudil (10 and 30 µM) and netarsudil (1 µM) for 24 to 72 h, respectively (*n* = 6), showing relative mRNA expression levels of *TJP1* (ZO-1), *TJP2* (ZO-2), *OCLN* (occludin), *CLDN2* (claudin 2), *GJA1* (connexin 43), *CDH2* (cadherin-2), *ITGA6* (integrin subunit alpha 6), and *ITGB1* (integrin subunit beta 1). Data are normalized to *GAPDH* (glyceraldehyde-3-phosphate dehydrogenase) and *HPRT1* (hypoxanthine phosphoribosyltransferase 1) and expressed as the means ±  SDs relative to controls set to 1 (* *p* < 0.05, ** *p* < 0.01,*** *p* < 0.001; unpaired *t*-test). (**B**) Expression of ZO-1 (green fluorescence) in pHCEnCs incubated without or with ripasudil (10 µM) and netarsudil (1 µM) for 48 h (DAPI nuclear counterstain in blue). (**C**) Effects of ROCK inhibitors on paracellular permeability and transcellular electrical resistance of confluent monolayers of pHCEnCs as assessed by toluidine blue vital staining, FITC–dextran permeability assay (*n* = 6) and TEER assay (*n* = 20) using fibroblasts as controls (data are expressed as the means ± SDs relative to controls; * *p* < 0.05; *** *p* < 0.001; **** *p* < 0.0001; Mann–Whitney *U* test).

**Figure 5 cells-14-00258-f005:**
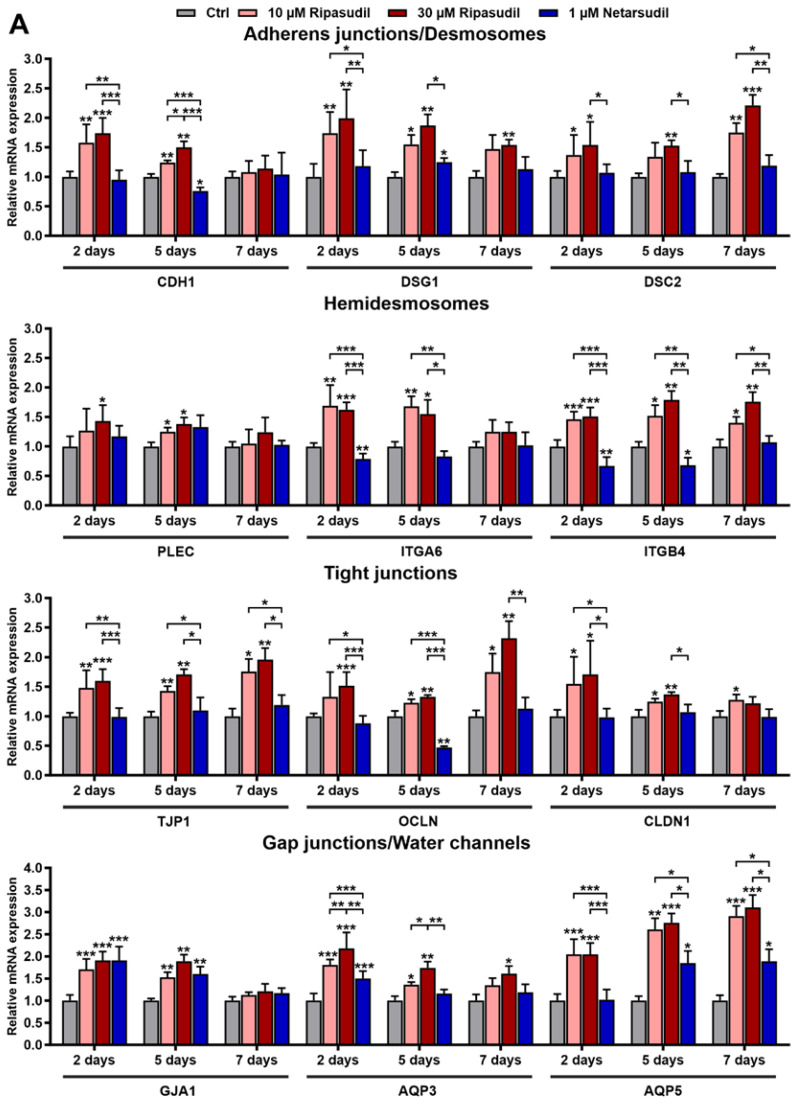
Effects of ripasudil and netarsudil on the regulation of corneal epithelial cell junctions. (**A**) Quantitative real-time PCR analysis of primary human corneal epithelial cells (pHCEpCs) treated without or with ripasudil (10 and 30 µM) and netarsudil (1 µM) for 2 to 7 days, respectively (*n* = 6), showing the relative mRNA expression levels of *CDH1* (cadherin-1), *DSG1* (desmoglein 1), *DSC2* (desmocollin 2), *PLEC* (plectin), *ITGA6* (integrin subunit alpha 6), *ITGB4* (integrin subunit beta 4), *TJP1* (ZO-1), *OCLN* (occludin), *CLDN1* (claudin 1), *GJA1* (connexin 43), *AQP3* (aquaporin 3), and *AQP5* (aquaporin 5). Data are normalized to *GAPDH* (glyceraldehyde-3-phosphate dehydrogenase) and *HPRT1* (hypoxanthine phosphoribosyltransferase 1) and expressed as the means ±  SDs relative to controls set to 1 (* *p* < 0.05, ** *p* < 0.01, *** *p* < 0.001; unpaired *t*-test). (**B**) Double immunofluorescence labeling of cadherin-1 (green fluorescence) and vimentin (red fluorescence) (top row), integrin α6 (green fluorescence) and phalloidin (red fluorescence) (second row from top), ZO-1 (green fluorescence) and vimentin (red fluorescence) (third row from top), and connexin 43 (green fluorescence) and cytokeratin pan (red fluorescence) (bottom row) in pHCEpCs incubated without or with ripasudil (10 and 30 µM) and netarsudil (1 µM) for 48 h (DAPI nuclear counterstain in blue). (**C**) Effects of the ROCK inhibitors on the paracellular permeability and transcellular electrical resistance of confluent monolayers of pHCEpCs as assessed by toluidine blue vital staining, FITC–dextran permeability assay (*n* = 6) and TEER assay (*n* = 8) using fibroblasts as controls (data are expressed as the means ± SDs relative to controls; * *p* < 0.05; ** *p* < 0.01; *** *p* < 0.001; **** *p* < 0.0001; Mann–Whitney *U* test).

**Figure 6 cells-14-00258-f006:**
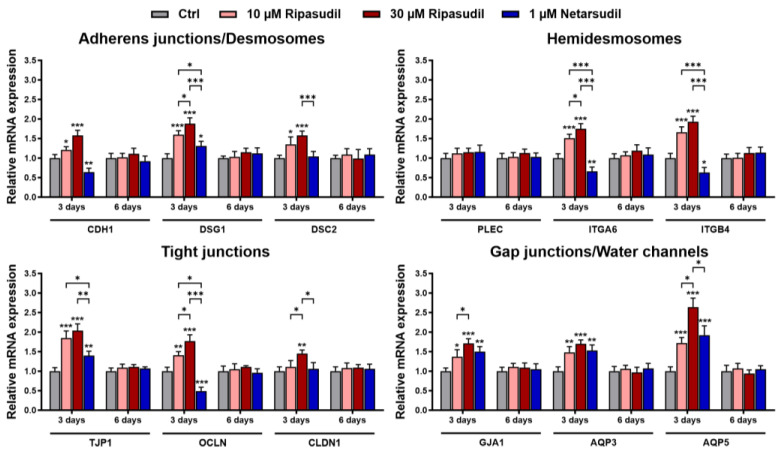
Effects of the discontinuation of ripasudil and netarsudil after 3 days on the regulation of corneal epithelial cell junctions. Quantitative real-time PCR analysis of primary human corneal epithelial cells (pHCEpCs) treated without or with ripasudil (10 and 30 µM) and netarsudil (1 µM) for 3 days and analyzed after 6 days (*n* = 6), showing the relative mRNA expression levels of *CDH1* (cadherin-1), *DSG1* (desmoglein 1), *DSC2* (desmocollin 2), *PLEC* (plectin), *ITGA6* (integrin subunit alpha 6), *ITGB4* (integrin subunit beta 4), *TJP1* (ZO-1), *OCLN* (occludin), *CLDN1* (claudin 1), *GJA1* (connexin 43), *AQP3* (aquaporin 3), and *AQP5* (aquaporin 5). Data are normalized to *GAPDH* (glyceraldehyde-3-phosphate dehydrogenase) and *HPRT1* (hypoxanthine phosphoribosyltransferase 1) and expressed as the means ±  SDs relative to controls set to 1 (* *p* < 0.05, ** *p* < 0.01, *** *p* < 0.001; unpaired *t*-test).

**Figure 7 cells-14-00258-f007:**
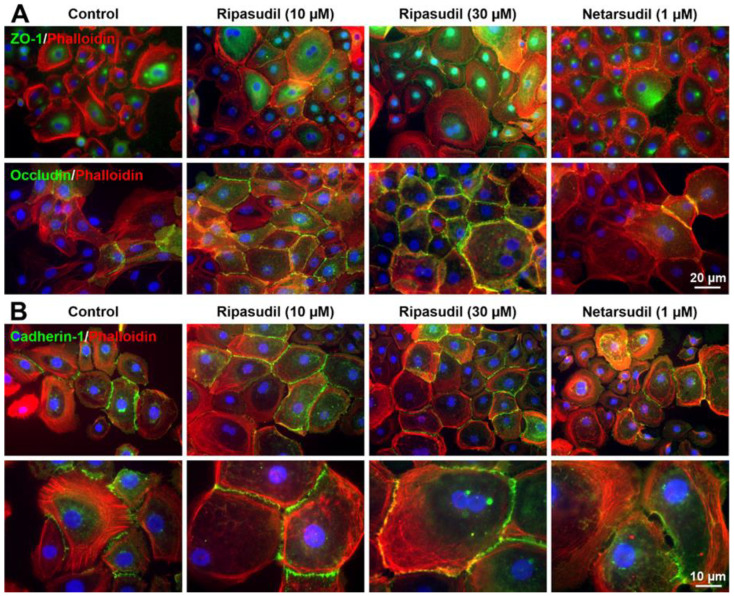
Effects of ripasudil and netarsudil on the actin cytoskeleton and formation of tight and adherens junctions in primary human corneal epithelial cells (pHCEpCs). (**A**) Double immunofluorescence labeling of ZO-1 (green fluorescence) and phalloidin (red fluorescence) (top row), and occludin (green fluorescence) and phalloidin (red fluorescence) (bottom row), in pHCEpCs incubated without or with ripasudil (10 and 30 µM) and netarsudil (1 µM) for 48 h. (**B**) Double immunofluorescence labeling of cadherin 1 (green fluorescence) and phalloidin (red fluorescence) in pHCEpCs (DAPI nuclear counterstain in blue).

## Data Availability

Data are contained within the article or Appendix A.

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
