# Peer review of "Drug- and Cell-Type-Specific Effects of ROCK Inhibitors as a Potential Cause of Reticular Corneal Epithelial Edema"

_cells, 2025, doi:10.3390/cells14040258_

Round 1
Reviewer 1 Report
Comments and Suggestions for Authors
This is a study on the mechanism underlying reticular corneal epithelial edema caused by two popular ROCK inhibitor-associated drugs using cultured primary human corneal endothelial and epithelial cells. The related cell models are interesting, but the study is not logically designed well and thereby the results do not convincingly clarify the concluded mechanism. I have following major and minor concerns:
Major concerns:
1. Ripasudil is a pure ROCK-I and ROCK-II inhibitor whereas netarsudil inhibits both ROCK and norepinephrine transporter. As indicated in the introduction, most reticular bullous epithelial edema complications were reported with netarsudil but NOT ripasudil treatment, the logical suspicious devil should be the effect of netarsudil on norepinephrine transporter. Therefore, the authors should design a study to compare the cellular effects of the pure ROCK inhibitor ripasudil to that of a pure norepinephrine transporter inhibitor.
2. The story starts with a single clinical case report on a congenital hereditary endothelial dystrophy (CHED) and secondary open-angle glaucoma patient who was treated with Roclanda that contains both 0.02% netarsudil and 0.005% latanoprost and had shown a reticular corneal epithelial edema. I do not understand why authors concluded here “likely due to netarsudil” not the prostaglandin analogue latanoprost unless they clearly cite references here to support their hypothesis?
3. Authors must detail their preparation of all primary corneal cells, not just cite a reference in the methods and morphologically and functionally characterize these cells in the results as cell models are set up to mimic their in vivo status.
Minor suggestions:
1. Page 2, please cite a reference for the sentence “However, reticular corneal epithelial edema usually resolved upon discontinuation of the medication.”
2. Page 2, better to mention Fuchs endothelial corneal dystrophy (FECD) is one of the congenital hereditary endothelial dystrophy (CHED) disorders so that readers would understand why select the clinical case to start in the first section of the results.
3. Page 3, please spell out the abbreviation “DMEK”.
4. Page 5, define latanoprost as a prostaglandin analogue when it first appear in the text.
5. Page 6, please cite a reference for “…with commonly used experimental concentrations of ripasudil (10 and 30 µM) and netarsudil (1 µM), corresponding to 0.0004% and 0.0012% ripasudil and 0.00005% netarsudil, for 72 hours.”
6. Page 8, Fig. 3B, the increase in Na+/K+-ATPase activity may also be due to the increase in cell numbers after ripasudil and netarsudil treatment.
7. Fig. 3B again, I do not understand why used vimentin here. Vimentin is a well-known marker for mesenchymal cells, does it mean that endothelial cell-Descemet membrane (EDM) isolated from FECD patients is contaminated with corneal stromal cells?
8. Page 9, CDH2 (N-cadherin) is a popular indicator for epithelial to mesenchymal transition, not a typical marker for endothelial cells.
9. Page 10, why Fig. 4B control ZO-1 image is different from Fig. 2A control ZO-1?
10. Page 10, Fig. 4C, TEER assay indicates almost no difference between Ctrl and Fibroblasts, thereby the endothelial to mesenchymal transition (EnMT) is obvious for cultured corneal endothelial cells (CEnC). This may indicate that both ripasudil and netarsudil inhibit EnMT once subject to culture conditions.
11. Page 11 Fig. 5B, again vimentin positive corneal epithelial cells here indicate an occurrence of EMT once subject to culture conditions, and that both ripasudil and netarsudil inhibit this EMT by increasing both ZO-1 and CDH1 expression.
Author Response
Point-by-point response
We would like to thank the reviewers and editors for their appreciation of our work, their thorough revision, and their constructive and critical comments. We have tried to address all issues raised as follows:
Reviewer #1
Major concerns:
- This is a study on the mechanism underlying reticular corneal epithelial edema caused by two popular ROCK inhibitor-associated drugs using cultured primary human corneal endothelial and epithelial cells. The related cell models are interesting, but the study is not logically designed well and thereby the results do not convincingly clarify the concluded mechanism. I have following major and minor concerns:
Ripasudil is a pure ROCK-I and ROCK-II inhibitor whereas netarsudil inhibits both ROCK and norepinephrine transporter. As indicated in the introduction, most reticular bullous epithelial edema complications were reported with netarsudil but NOT ripasudil treatment, the logical suspicious devil should be the effect of netarsudil on norepinephrine transporter. Therefore, the authors should design a study to compare the cellular effects of the pure ROCK inhibitor ripasudil to that of a pure norepinephrine transporter inhibitor.
The rationale of the study was to comparatively analyze the differential effects of two ROCK inhibitors in clinical use on corneal cells by testing existing hypotheses on the cause of reticular epithelial edema mainly (but not exclusively) observed after treatment with netarsudil. Therefore, we had to include both ROCK inhibitor drugs in our study instead of including a pure norepinephrine transporter inhibitor, such as fluoxetine and other neurochemical drugs, which have not been reported to induce corneal epithelial edema.
- The story starts with a single clinical case report on a congenital hereditary endothelial dystrophy (CHED) and secondary open-angle glaucoma patient who was treated with Roclanda that contains both 0.02% netarsudil and 0.005% latanoprost and had shown a reticular corneal epithelial edema. I do not understand why authors concluded here “likely due to netarsudil” not the prostaglandin analogue latanoprost unless they clearly cite references here to support their hypothesis?
The case report has been included in the manuscript to provide the non-specialized reader of „Cells“ with the clinical appearance of reticular corneal epithelial edema. In Europe, only a fixed-dose combination of netarsudil 0.02% and latanoprost 0.005% has been approved for treatment of glaucoma. We reported the first case of reticular epithelial edema following fixed combination of netarsudil and latanoprost (Roclanda®) for IOP control. Since reticular corneal epithelial edema has been repeatedly reported after netarsudil use, but never after latanoprost use, we concluded that it developed „likely due to netarsudil“ confirming previous reports.
- Authors must detail their preparation of all primary corneal cells, not just cite a reference in the methods and morphologically and functionally characterize these cells in the results as cell models are set up to mimic their in vivo status.
Primary human corneal endothelial cells (pHCEnC) were obtained from Celprogen (Torrance, CA) and positively tested for expression of endothelial cell markers, such as Na+/K+-ATPase, ZO-1 and N-cadherin both on the transcript and protein level. Functional characterization further included verification of Na/K-ATPase activity (Fig. 3B).
Primary human corneolimbal epithelial cells (pHCEpC) were prepared from organ-cultured donor corneas as has been described in about 20 previous papersof our group including the cited reference. Epithelial cells were characterized by positive expression of epithelial markers, including CDH-1, DSG1, DSC2, ITGA6, ITGB4, GJA1, both on the transcript and protein levels (Fig. 4A,B). However, we have additionally analyzed expression of corneal epithelium specific markers, KRT3, KRT12 and p63 alpha by RT-PCR and immunohistochemistry. These data are decribed in the Methods and are shown in Supplemental Figure S3.
Minor suggestions:
- Page 2, please cite a reference for the sentence “However, reticular corneal epithelial edema usually resolved upon discontinuation of the medication.”
References have been included.
- Page 2, better to mention Fuchs endothelial corneal dystrophy (FECD) is one of the congenital hereditary endothelial dystrophy (CHED) disorders so that readers would understand why select the clinical case to start in the first section of the results.
Unfortunately, FECD cannot be classified as a congenital hereditary endothelial dystrophy, because both disorders are distinctly different clinical entities. As mentioned above, the clinical case has been included in the mansuscript to illustrate the bullous corneal epithelial alterations, which are addressed in the present study.
- Page 3, please spell out the abbreviation “DMEK”.
DMEK has been explained on page 3.
- Page 5, define latanoprost as a prostaglandin analogue when it first appear in the text.
Latanoprost has been defined on page 2.
- Page 6, please cite a reference for “…with commonly used experimental concentrations of ripasudil (10 and 30 µM) and netarsudil (1 µM), corresponding to 0.0004% and 0.0012% ripasudil and 0.00005% netarsudil, for 72 hours.”
References have been provided.
- Page 8, Fig. 3B, the increase in Na+/K+-ATPase activity may also be due to the increase in cell numbers after ripasudil and netarsudil treatment.
We agree with the reviewer that the increase in enzymatic activity compared to controls may be partly due to the increase in cell numbers after treatment. However, Fig. 3A also shows an increase in ATP1A1 expression upon ROCK inhibition after normalization to house keeping genes. In addition, the main aspect of this experiment is the comparison of effects of ripasudil and netarsudil on endothelial cells.
- 3B again, I do not understand why used vimentin here. Vimentin is a well-known marker for mesenchymal cells, does it mean that endothelial cell-Descemet membrane (EDM) isolated from FECD patients is contaminated with corneal stromal cells?
We understand the reviewer’s concerns about positive vimentin staining of corneal endothelial cells of Descemet membrane specimens. However, endothelial cells from FECD patients have been shown to express vimentin as a sign of EnMT, which is characteristic for this disorder (e.g. Hidayat AA, Cockerham GC. Epithelial metaplasia of the corneal endothelium in Fuchs endothelial dystrophy. Cornea. 2006 Sep;25(8):956-9). Moreover, most cell types start to express vimentin in cell or tissue culture (e.g. Pieper FR, Van de Klundert FA, Raats JM, Henderik JB, Schaart G, Ramaekers FC, Bloemendal H. Regulation of vimentin expression in cultured epithelial cells. Eur J Biochem. 1992 Dec 1;210(2):509-19).
- Page 9, CDH2 (N-cadherin) is a popular indicator for epithelial to mesenchymal transition, not a typical marker for endothelial cells.
In contrast to epithelial cells, where the change in expression from CDH1 to CDH2 marks the process of EMT, corneal endothelial cells constitutively express CDH2 as a component of adherens junctions (e.g. Vassilev VS, Mandai M, Yonemura S, Takeichi M. Loss of N-cadherin from the endothelium causes stromal edema and epithelial dysgenesis in the mouse cornea. Invest Ophthalmol Vis Sci. 2012 Oct 17;53(11):7183-93).
- Page 10, why Fig. 4B control ZO-1 image is different from Fig. 2A control ZO-1?
Fig. 4B and Fig. 2A illustrate different experiments. Fig. 2A shows characterization of pHCEnC, which were stained for endothelial markers immediately upon confluency. Fig. 4B shows an experimental approach using pHCEnC, which were incubated with or without ROCK inhibitors for additional 48 hours after confluency. Most importantly, however, both images reveal typical membrane staining for ZO-1.
- Page 10, Fig. 4C, TEER assay indicates almost no difference between Ctrl and Fibroblasts, thereby the endothelial to mesenchymal transition (EnMT) is obvious for cultured corneal endothelial cells (CEnC). This may indicate that both ripasudil and netarsudil inhibit EnMT once subject to culture conditions.
Although fibroblasts exhibit a significantly (p<0.05) reduced TEER compared to controls, it is quite likely that cultured pHCEnCs undergo some degree of EnMT during cultivation. Most importantly, however, they still express ths functional markers ZO-1 and Na/K-ATPase. The reviewer correctly states that ROCK inhibitors may suppress EnMT, as shown in our previous study for ripasudil (Schlötzer-Schrehardt U et al. Potential Functional Restoration of Corneal Endothelial Cells in Fuchs Endothelial Corneal Dystrophy by ROCK Inhibitor (Ripasudil). Am J Ophthalmol. 2021 Apr;224:185-199).
- Page 11 Fig. 5B, again vimentin positive corneal epithelial cells here indicate an occurrence of EMT once subject to culture conditions, and that both ripasudil and netarsudil inhibit this EMT by increasing both ZO-1 and CDH1 expression.
It is correct that vimentin expression by cultured corneal epithelial cells may indicate initiation of the reversible EMT program (e.g. Castro-Muñozledo F, Meza-Aguilar DG, Domínguez-Castillo R, Hernández-Zequinely V, Sánchez-Guzmán E. Vimentin as a Marker of Early Differentiating, Highly Motile Corneal Epithelial Cells. J Cell Physiol. 2017 Apr;232(4):818-830), and it is reasonable to assume that this can be reversed by ROCK inhibitors.
Reviewer 2 Report
Comments and Suggestions for Authors
This manuscript entitled "Drug- and cell type-specific effects of ROCK inhibitors as a potential cause of reticular corneal epithelial edema" investigated the effects of ripasudil and netarsudil on corneal endothelial and epithelial function in vitro. I have some comments below.
1. Specify the informed consent for the case report.
2. Why two concentrations of lipasudil were used, but one concentration of netarsudil?
3. For gene expression, not only qPCR but also RNAseq should be used to understand the interactions and regulatory networks between specific gene groups.
4. Describe limitation of this study in Discussion section.
Author Response
- Specify the informed consent for the case report.
Informed consent has been provided.
- Why two concentrations of lipasudil were used, but one concentration of netarsudil?
The two concentrations of ripasudil are based on findings of our previous study (Am J Ophthalmol. 2021 Apr;224:185-199): An initial dose-response experiment (10, 30 and 100 µM) showed the most pronounced effects of ripasudil on gene expression changes of EDM specimens obtained from FECD patients and normal donor corneas at a concentration of 30 μM. Similar dose-response experiments were performed with cultured endothelial cells, which responded already at a concentration of 10 µM of rispasudil. Regarding netarsudil, we referred to the literature showing a concentration of 1 µM to be most effective in cell culture experiments (e.g., trabecular endothelial cells). A gross morphological analysis of corneal endothelial/epithelial cells using different concentrations of netarsudil (0.5-10 µM) showed cell death at 10 µM confirming published studies.
- For gene expression, not only qPCR but also RNAseq should be used to understand the interactions and regulatory networks between specific gene groups.
We completely agree with the reviewer. In fact, we have performed a RNAseq analysis of ripasudil-treated endothelial cells, which will be published in a follow-up study and which could be expanded to netarsudil-treated cells. Since the main goal of this study was to test existing hypotheses on netarsudil-induced development of epithelial bullae, we considered a selective approach with appropriate markers sufficient to prove/ disprove hypotheses on barrier function, etc. We do hope to satisfy the reviewer with this limitation, which has been addressed in the Discussion.
- Describe limitation of this study in Discussion section.
A paragraph describing the limitations has been included in the Discussion as follows:
“The study has some limitations. A major limitation is that only a 2D in vitro model of corneal epithelial cells has been used to assess the effects of ROCK inhibitors on epithelial barrier function. To better recapitulate the multilayered in vivo tissue architecture, 3D in vitro models of the corneal epithelium should be used in future approaches, such as primary human corneal epithelial cells forming stratified epithelia at the air-liquid interface (Kaluzhny et al. 2018). Using such models will substantiate and further improve analyses of drug effects on cellular interactions, barrier disruptions and permeability function. Another limitation of the present study is the selective approach of gene expression alterations using appropriate markers of endothelial pump and barrier as well as epithelial barrier function. While this limited approach may have been sufficient to test existing hypotheses on the development of epithelial bullae in vitro, future studies should perform an unbiased transcriptomic analysis to provide further mechanistic insights into drug-induced cellular and molecular effects and regulatory networks between differentially expressed genes.”
Round 2
Reviewer 1 Report
Comments and Suggestions for Authors
None